# Profiling the Atopic Dermatitis Epidermal Transcriptome by Tape Stripping and BRB-seq

**DOI:** 10.3390/ijms23116140

**Published:** 2022-05-30

**Authors:** Tu Hu, Tanja Todberg, Daniel Andersen, Niels Banhos Danneskiold-Samsøe, Sofie Boesgaard Neestrup Hansen, Karsten Kristiansen, David Adrian Ewald, Susanne Brix, Joel Correa da Rosa, Ilka Hoof, Lone Skov, Thomas Litman

**Affiliations:** 1Bioinformatics, Molecular Biomedicine, LEO Pharma A/S, 2750 Ballerup, Denmark; uyhdk@leo-pharma.com (T.H.); ddadk@leo-pharma.com (D.A.E.); ilhoo@leo-pharma.com (I.H.); 2Department of Immunology and Microbiology, University of Copenhagen, 2200 Copenhagen, Denmark; 3Department of Dermatology and Allergy, Copenhagen University Hospital—Herlev and Gentofte, 2900 Copenhagen, Denmark; tanja.todberg@regionh.dk (T.T.); lone.skov.02@regionh.dk (L.S.); 4Department of Clinical Medicine, University of Copenhagen, 2200 Copenhagen, Denmark; 5Department of Biotechnology and Biomedicine, Technical University of Denmark, 2800 Lyngby, Denmark; daan@dtu.dk (D.A.); sbrix@dtu.dk (S.B.); 6Department of Biology, University of Copenhagen, 2100 Copenhagen, Denmark; nds@bio.ku.dk (N.B.D.-S.); sofie.neestrup@bio.ku.dk (S.B.N.H.); kk@bio.ku.dk (K.K.); 7Laboratory of Inflammatory Skin Diseases, Icahn School of Medicine at Mount Sinai, New York, NY 10029, USA; joel.correadarosa@mssm.edu

**Keywords:** tape stripping, atopic dermatitis, epidermis, transcriptome, BRB-seq

## Abstract

Tape stripping is a non-invasive skin sampling technique, which has recently gained use for the study of the transcriptome of atopic dermatitis (AD), a common inflammatory skin disorder characterized by a defective epidermal barrier and perturbated immune response. Here, we performed BRB-seq—a low cost, multiplex-based, transcriptomic profiling technique—on tape-stripped skin from 30 AD patients and 30 healthy controls to evaluate the methods’ ability to assess the epidermal AD transcriptome. An AD signature consisting of 91 differentially expressed genes, specific for skin barrier and inflammatory response, was identified. The gene expression in the outermost layers, stratum corneum and stratum granulosum, of the skin showed highest correlation between tape-stripped skin and matched full-thickness punch biopsies. However, we observed that low and highly variable transcript counts, probably due to low RNA yield and RNA degradation in the tape-stripped skin samples, were a limiting factor for epidermal transcriptome profiling as compared to punch biopsies. We conclude that deep BRB-seq of tape-stripped skin is needed to counteract large between-sample RNA yield variation and highly zero-inflated data in order to apply this protocol for population-wide screening of the epidermal transcriptome in inflammatory skin diseases.

## 1. Introduction

Atopic dermatitis (AD) is a common inflammatory skin disorder, which affects up to 20% of children and 10% of the adult population [1]. The pathogenesis of AD is characterized by a defective epidermal barrier function and a perturbed type 2 immune response [1,2].

To date, transcriptomic profiling of AD has largely relied on punch biopsies, which are able to collect full-thickness skin samples, thus providing information on all layers of the skin [3,4]. However, such invasive sampling can lead to scarring and risk of infections, which is why alternative and non-invasive techniques are desirable [5].

Tape stripping is such a non-invasive skin sampling technique that applies adhesive tapes to obtain molecular information on mRNA [6], proteins [7], lipids [8], and microbiome [9] from the skin. Because the technique is easily applicable, it is particularly useful for sampling large patient cohorts and for pediatric patients, from whom an epidermal transcriptomic profile may lead to discovery of biomarkers [10,11]. The RNA yield from tape-stripped skin is low, which makes it unfeasible to assess the level of RNA degradation, or to normalize input for cDNA synthesis and amplification. Thus, methods that are less sensitive to RNA quality and input amount are desirable [12].

RNA-seq has been used widely for transcriptomic profiling of AD. However, the relatively high cost of standard RNA-seq prohibits it to replace qPCR in routine analysis. Bulk RNA Barcoding and sequencing (BRB-seq) provides low cost (~20 USD per sample, including library preparation and sequencing) high-throughput transcriptomics profiling, with high tolerance for low RNA quality [12]. While BRB-seq does not provide information on splice variants, and short reads (60–70 bp) can lead to ambiguous mapping as compared to longer read (100–150 bp) RNA-seq, these appear as acceptable methodological trade-offs for cost-efficient, large-scale sequencing.

Tape stripping is increasingly used for non-invasive skin sampling and subsequent transcriptomic profiling, which has relied on qPCR or deep sequencing. In a study on pediatric AD patients, qPCR analysis of tape-stripped skin confirmed skin barrier abnormalities and activation of type 2 immune response in early-onset AD [6]. Sølberg et al. found that the stratum corneum transcriptome of AD derived from tape-stripped skin and matched full-thickness biopsies was comparable, and identified markers for Th1, Th2, Th17, Th22, and keratinocyte-induced signaling pathways [13]. In another study, where tape-stripped AD skin was profiled by RNA-seq, Dyjack et al. were able to identify type 2 inflammation markers for patient stratification [14].

Our study utilized the Gentofte atopic dermatitis (GENAD) cohort [15], with the aim of evaluating non-invasive tape stripping coupled with low cost BRB-seq for its applicability in clinical dermatology, specifically, for the characterization of the epidermal transcriptome in AD.

## 2. Results

### 2.1. Characteristics of Subjects

A total of 194 samples were obtained by applying tape stripping on 30 patients with mild-to-moderate AD, and 30 age and gender matched healthy controls: 66 samples from lesional (LS), 66 from non-lesional (NL), and 62 from healthy control (HC) skin. Subject and sample metadata are available in Appendix A.

### 2.2. BRB-seq and Data Quality Control

The extracted RNA was subjected to BRB-seq. We generated expression profiles of 194 samples. In general, we obtained relatively small library sizes for most samples (median: 33,421 read counts, range: 361–7,867,531).

After applying our 4-point data quality control (QC) and filtering scheme (shown in materials and methods), 132 samples remained for further analysis. A total of 13 LS, 22 NL, and 27 HC samples failed to pass the QC. More than half of the filtered samples had fewer than 200 genes detected and a negligible epidermal differentiation complex signal.

For the samples that passed QC filtering, we observed a large variation in total read count, suggesting a corresponding large deviation in input RNA, which could be due to either substantial differences in RNA yield, or RNA degradation. As shown in Figure 1a, the left-most cluster is characterized by higher overall expression of almost all genes compared to the remaining samples. In the corresponding PCA plot (Figure 1b), samples were separated by the first principal component, which explains 72% of the variance and can be attributed to the sequencing depth (counts <35,000).

To understand the variables affecting total read count, we performed linear mixed effect modeling. We found tissue state to have the strongest effect on total count (Figure 2a), showing an increasing trend for HC < NL < LS (Figure 2a). We also observed seasonal albeit non-significant variation in total counts (Figure 2b), as well as in gender, magnitude of hair signature (as estimated by KRTAP genes), and anatomic region of stripping.

### 2.3. Detection of Skin Specific Genes by Tape Stripping

We detected expression of 2310 genes (with > 1 count in 80% of samples) in total after filtering out non-protein coding and mitochondrial genes. The detected transcripts were enriched for the gene ontology terms of cornified envelope (q < 0.001, Odds Ratio = 57.0), and epidermis (q < 0.001, Odds Ratio = 6.43).

The following genes were found to be highly expressed across all tape strip skin samples: S100A8, S100A7, B2M, LORICRIN, FLG, SPRR2E, SPRR2EG, LYZ, CTNNB1, KRTAP3-2, and C1orf56, all of which have been reported to be enriched in skin [16,17].

We were also able to detect the expression of genes previously reported to be highly abundant in tape-stripped skin [14], including KRT10, KRT14, KRTAP1-5, KRT5, KRTAP4-9, FLG2, KRT86, KRT17, and ACTG1.

### 2.4. Benchmarking Data Normalization and Differential Expression Testing Methods

The large variation in read counts between samples and therefore, also, the level of zero-inflation (data points being zeros) in the data, could potentially skew differential expression (DE) analysis. Therefore, we benchmarked 12 data handling methods (transformation and DE testing combination) as listed in Table 1, with the full results shown online (bit.ly/sts-benchmark, accessed on 30 May 2022).

To determine the accuracy of the benchmarked methods, we considered the differentially expressed genes (DEGs) obtained from corresponding, full-thickness skin biopsies (that is: biopsies obtained from the same subject from matched skin sites) previously reported by Hu et al. [15] as the gold standard. We found that no existing data handling method could capture all DEGs. The data handling methods in the benchmark showed an overall low accuracy and low overlap in identified DEGs (Figure 3). For example, the commonly used DESeq2 algorithm had an accuracy of 1.02% for this data. Interestingly, the mitochondrial gene normalization method, although having a high false positive rate, could capture nine additional DEGs compared to all other methods tested (Figure 3). The benchmark results suggest that the trimmed mean of M value (TMM) transformed data together with voom or NOIseq (a non-parametric testing method) have higher accuracy in identifying DEGs from noisy, zero-inflated RNA-seq data.

### 2.5. Atopic Dermatitis Gene Expression Signature Obtained by Tape Stripping

We generated an AD gene expression signature by aggregating an ensemble of DEGs that were accurately identified by all 12 methods. The AD gene expression signature comprised 91 genes, of which 81 were up- and 10 were downregulated in LS compared to NL and HC skin.

Most of the AD signature genes are skin barrier, or skin barrier modulator genes, including FLG2, LORICRIN, KRT31, LCE3A/C/D/E, S100A2/7/7A/8/9, CSTA, APOBEC3A, CNFN, and SPRR1B/2A/2B/2D/2E/2F/2G. We were also able to detect genes associated with inflammatory and antimicrobial function (ADAM19, BCL2A1, and CLEC7A), cytokine and chemokine signaling (CCL22, CCL4L2, and IL4R), and G-protein signaling (RGS1) (Figure 4). However, it is worth noting the large intra-group variation, as shown in Appendix A.

We compared our AD gene expression signature to those obtained from two other studies where deep RNA-seq was used for profiling the epidermal transcriptome from tape-stripped skin [13,14]. As shown in Figure 5a, we found little overlap between AD signature genes from the three studies. In contrast, pathway enrichment analysis showed good agreement between the most enriched pathways, namely those related to epidermal structure and inflammation (Figure 5b). The Dyjack 2018 data demonstrated a stronger inflammatory response, which probably can be ascribed to the more severe AD in that study. Thus, the limited overlap between AD signature genes could be due to both variation in disease severity and in tape stripping methodology.

We were unable to detect seasonal variation in the epidermal transcriptome, which is in line with our results obtained from matched full-thickness skin biopsies [15].

### 2.6. Differences between Tape Strip and Punch Biopsy-Derived Atopic Dermatitis Gene Signatures

We compared the expression of the 2310 genes that were detected in tape-stripped skin samples with that of matched full-thickness punch biopsies. We found a weak to moderate correlation between the two sampling (and analytical) methods, with a median Spearman correlation coefficient of 0.13.

We also observed that tape-stripped samples collected from LS sites had significantly higher correlation with matched full-thickness punch biopsies, in comparison with NL (*p* = 0.023) and HC (*p* < 0.001) (Figure 6a), which is probably due to higher RNA yield in LS than in NL and HC. We also found that a high correlation between tape-stripped and full-thickness punch biopsy samples was significantly associated with high total transcript counts from tape-stripped samples in all skin types (*p* < 0.001) (Figure 6b). A comparison of gene expression fold change between LS and NL samples did not show better correlation between the two methods than the direct counts comparison.

As shown in Figure 7, we found the highest correlation between gene expression detected in tape-stripped skin and matched full-thickness punch biopsies in the two outermost epidermal layers of the skin, the stratum corneum and stratum granulosum. In particular, the expression of epidermal differentiation complex genes (*LCE3A*, *S100B*, and *S100A7*) is highly correlated. In contrast, expression of stratum spinosum and stratum basale specific genes demonstrated no significant correlation between tape-stripped and full-thickness punch biopsies.

## 3. Discussion

Tape stripping is an emerging, non-invasive, and easily applicable skin sampling technique, which has been used to study mRNA [6], proteins [7], lipids [8], and microbiome [9] on the skin surface, especially in pediatric patients [6,24].

However, the reproducibility and quality of tape stripping has caused some controversy, which can be attributed to lack of a standardized sampling method as well as the potential leakage of serum proteins on the tapes in diseases with a disrupted epidermal barrier [25]. In addition, human skin is rich in RNAses, which is why molecular diagnostics based on gene expression in tape-stripped samples should be thoroughly validated to avoid negative results due to RNA degradation [25].

In the present study, the tape strips were stored at −80 °C without RNA stabilizing preservatives, and therefore, we decided to use BRB-seq, an RNA-seq technique with reported high tolerance for low RNA quality [12]. Thus, despite low RNA yield and quality from the tape-stripped skin samples, we were still able to detect a meaningful skin and AD specific signal, arising from the stratum corneum and stratum granulosum layer of the epidermis. The AD specific gene signature identified in our study could confirm 72% (91/126) of the DEGs reported by Sølberg et al., who profiled AD by tape stripping followed by deep RNA sequencing [13]. Thus, our results agree with and extend previous studies on the practicability of tape stripping in studying the stratum corneum transcriptome in AD [13,24].

Our results suggest that tape stripping of skin poses both technical and analytical challenges to the identification of epidermal mRNA biomarkers. Because of the low and highly variable transcript counts, probably due to low RNA yield and RNA degradation, deep sequencing is required to obtain a sufficiently high signal-to-noise ratio. Furthermore, BRB-seq applies the early pooling of the samples, which makes it hard to normalize for the RNA input in one BRB batch, especially when the RNA concentration between samples varies considerably. Finally, our benchmark results demonstrate that the current data handling techniques are not optimal for handling highly zero-inflated data with large variation in total transcript counts. A recent study suggests that two popular DE testing methods, edgeR and DESeq2, generate exaggerated false positive findings for large (>8 for each condition) sample sizes [26]. Our benchmark results confirmed the abovementioned problem in tape-stripped skin samples.

To increase the usability of tape stripping in dermatological research and diagnosis, one could improve the technique, both by minimizing sample variation and by maximizing RNA yield and quality. This may be accomplished by standardizing the sampling conditions, both technical parameters such as controlling the pressure, duration, and speed of tape removal, as well as biological factors, including anatomical site, hydration and stretching of the skin, as proposed by Hughes et al. [27]. To avoid RNA degradation during storage, the addition of RNA preservatives, such as RNAlater, should be considered. On the other hand, dry storage of non-conserved tape strips at room temperature for up to three days followed by deep sequencing (average depth of 90 M reads per sample) has been used successfully for transcriptomic profiling of the stratum corneum in AD [13] as well as in hand eczema [28].

## 4. Materials and Methods

### 4.1. Collection of Tape Stripping and Sample Processing

We recruited 30 AD patients and 30 age and gender matched HC from the outpatient clinic at Gentofte Hospital between April 2018 and November 2019. The AD group was characterized by mild-to-moderate severity, with mean eczema area and severity index (EASI) 4.0 and mean SCORing Atopic Dermatitis (SCORAD) 32.6. The patients were restricted from systemic anti-inflammatory treatment >4 weeks and from local treatment >2 weeks before the visit. Patients with contact allergies, malignancies, infections, and patients receiving other immune modulative treatment were excluded. The specific patient details can be found in Appendix A. In total, 194 samples were collected by tape stripping: 66 from LS, 66 from NL, and 62 from HC skin. Tape-stripped skin samples were obtained by attaching adhesive tape strips (diameter 22 mm, Disc D-Squame, Monaderm, Monaco, France) to the skin for 10 s with a standardized pressure (225 g cm^−2^) using the D-Squame pressurizer D500 (CuDerm, Dallas, TX, USA).

Each sample consisted of 8 consecutive D-squame tapes, each of which were placed in individual Eppendorf tubes and transferred to −80 °C immediately after collection. Total RNA was extracted from the 5th to 8th tape (RNAqueous, Life technologies) followed by digestion with DNase I according to the manufacturer’s instructions.

### 4.2. BRB-seq and Data Processing

Multiplexed cDNA libraries were generated using the BRB-seq workflow [12]. First-strand synthesis and barcoding of cDNA was performed by the addition of 5.75 µL DNase-free water, 1 µL of 10 µM oligo-dT primers and 1 µL of dNTP (10 mM) to 6 µL of RNA followed by 5 min incubation at 65 °C. This was followed by reverse transcription by the addition of 2 µL of DTT (Invitrogen), 0.25 µL of SuperScript II enzyme (Invitrogen), 4 µL of SuperScript II 5X Buffer (Invitrogen, Waltham, MA, USA), and 1 µL of 10 µM template-switching oligo (Integrated DNA Technologies, Coralville, IA, USA) in TE buffer (Invitrogen, AM9858). The mix was then incubated at 42 °C for 50 min, followed by inactivation at 70 °C for 15 min in a thermal cycler. To pool and concentrate the samples, the five pools of barcoded cDNA were then purified using the DNA Clean & Concentrator kit (Zymo Research, Irvine, CA, USA) according to the manufacturer’s instructions. This was immediately followed by exonuclease treatment by the addition of E. coli Exonuclease I (NEB, #M0293) and incubation at 37 °C for 30 min followed by enzyme inactivation at 80 °C for 20 min, resulting in 20 µL of pooled and exonucleated cDNA. The second-strand cDNA synthesis was performed by the addition of 1 µL of 10 µM LA_oligo (Integrated DNA Technologies), 1 µL of 0.2 mM dNTP, 1 µL of Advantage 2 polymerase mix (Clontech, #639206), 5 µL of Advantage 2 PCR buffer (Clontech, #639206) and 22 µL of DNase-free water, and then PCR amplification (95 °C, 1 min, then 10 cycles of 95 °C for 15 s, 65 °C for 30 s, 68 °C for 6 min.; and finally 72 °C for 10 min). The five cDNA pools were then purified using AMPure XP magnetic beads (Beckman Coulter, #A63881) according to the manufacturer’s instructions. Then, 1 ng of purified cDNA was tagmented (Illumina, Nextera XT DNA library Prep Kit) by adding 10 µL of Nextera ‘Tagment DNA buffer’ and 5 μL of ‘Amplicon Tagment mix’, then incubated at 55 °C for 5 min, and finally the Tn5 transposase was dissociated by the addition of 5 µL of ‘NT Buffer’ followed by incubation at room temperature for 5 min. Then, the indexing of tagmented DNA was performed by the addition of 1 µL of 10 µM P5 BRB oligo (Integrated DNA Technologies), 1 µL of 10 µM Nextera N70X oligos, 8 µL of RNase/DNase-free water and 15 µL of Nextera ‘NPM PCR mastermix’ and amplified by PCR (first 72 °C 3 min, then 95 °C 30 s; then 22 cycles of: 95 °C 10 s, 55 °C 30 s, 72 °C 30 s, and finally 72 °C for 5 min). The five indexed libraries were then purified using AMPure XP magnetic beads (Beckman Coulter, #A63881) according to the manufacturer’s instructions. Finally, the library quality was assessed by evaluation using a high sensitivity bioanalyzer chip (Agilent, Santa Clara, CA, USA) and sequenced on a NextSeq 500 instrument (Illumina, San Diego, CA, USA). All primers are similar to those used in [12].

In total, 194 samples were sequenced in five BRB batches generating a total of 343 M reads. Batch effect was examined, and not observed. BRB-seq data was mapped by aligning reverse reads to the human reference genome (GRCh38.105) using STAR(2.7.9a) [29], then de-multiplexing by forward read barcoding using the BRB-seq tool 1.6.0, resulting in an average mapping rate of 70%.

### 4.3. Four-Point Data Quality Control Scheme

We evaluated sample quality by a 4-point data QC scheme. Each sample scored 1 point for fulfilling each of the following criteria: (1) high sensitivity of detection (detecting >200 genes, at least 1 count); (2) high EDC signal, as measured by the sum of FLG, FLG2, LORICRIN, LCE1A, S100A7, S100A8, SPRR2E, SPRR1B, and IVL transcript counts higher than 100; (3) EDC gene counts comprise >5% of total counts; (4) mitochondrial genes comprise <15% of total counts. The samples with ≥3-point were kept for analysis.

### 4.4. Data Analysis

Data analysis was performed in R 4.1.2 (R foundation). EnrichR [30] was applied for enrichment testing, using the Human Gene Atlas [31] and Gene Ontology [32] as the reference gene set. A linear mixed effect model was fitted by lme4 [33], and the statistical significance was calculated by LmerTest [34].

RLE, TMM, and TMMwsp transformation were performed using edgeR [19]. Voom-trend was estimated by voom using TMM transformed data [21]. ZINB-Wave transformation was performed by the modeling sequencing depth as a sample-level covariate. Mitochondrial gene set normalization was performed by calculating the ratio between each transcript count and the sum of MT-ATP6/8, MT-CO1/2/3, MT-CYB, and MT-ND2/3/4/4L/5/6 transcript counts. When testing DEG between the tissue state, the subject was eliminated as a factor in paired analysis (LS vs. NL), and gender was eliminated as a factor in unpaired analysis (LS vs. HC and NL vs. HC). NOIseq was performed on TMM transformed data. The cutoff for DE analysis is documented in Table 1 for each benchmarked method.

The gene expression of biological replicates from punch biopsies was averaged when calculating the Spearman correlation coefficients between tape-stripped samples with matched full-thickness punch biopsies. We have documented the data analysis pipeline in R markdown files, available at the GitHub repository (github.com/tuhulab/multiomics-ad-sts, accessed on 30 May 2022).

## 5. Conclusions

Tape stripping of skin is a promising and easily applicable technology, and is particularly useful for sampling pediatric patients. We envisage that tape stripping, combined with affordable deep BRB-seq, has the potential for population-wide screening of the epidermal transcriptome in inflammatory skin diseases. However, because the reproducibility and across-study comparability of transcriptomic data generated by tape stripping is low, the technique needs to be improved by further studies investigating standardized sampling, sample, and data handling methods.

## Figures and Tables

**Figure 1 ijms-23-06140-f001:**
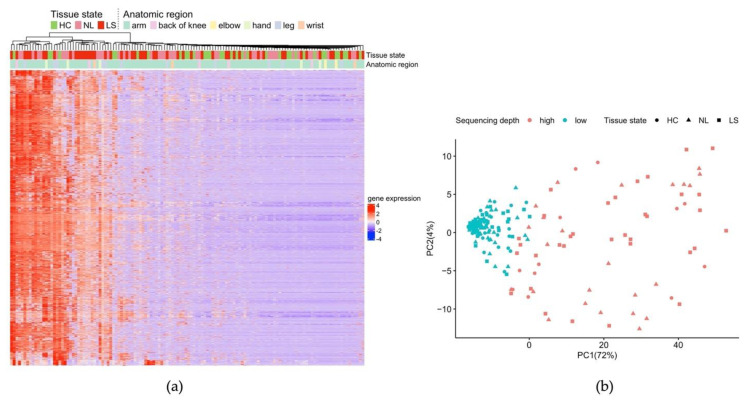
Overview of skin tape stripping BRB-seq data. (**a**) Heatmap and unsupervised hierarchical clustering based on 2310 expressed genes (rows) across 132 samples (columns). Counts are z-scaled. The anatomical region of sampling (arm: 106, leg: 11, hand: 5, back of knee: 5, wrist: 3, and elbow: 2) as well as tissue state are indicated above the plot; (**b**) corresponding sample PCA plot based on the same 2310 genes colored according to sequencing depth (high: counts ≥35,000).

**Figure 2 ijms-23-06140-f002:**
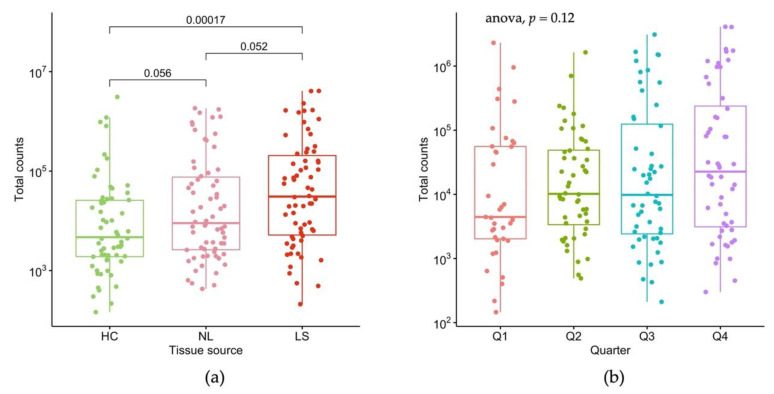
Total count variation. (**a**) The boxplot shows total count having an increasing trend in HC < NL < LS. *P*-values are shown above the contrasts; (**b**) the boxplot shows that RNA yield varies in different quarters of the year, but the differences did not reach statistical significance (*p* = 0.12).

**Figure 3 ijms-23-06140-f003:**
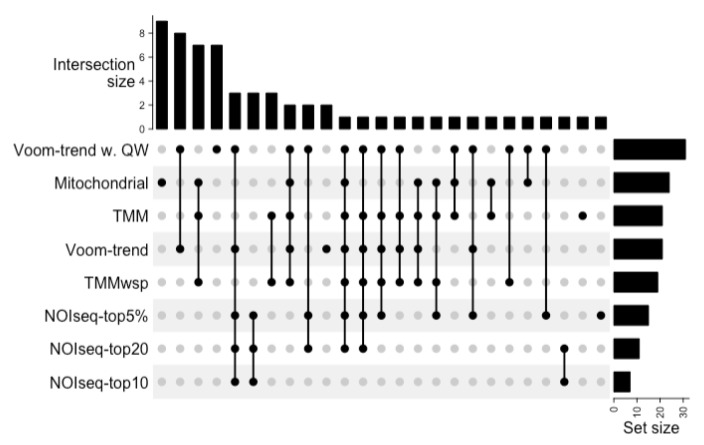
Upset plot showing the intersection between differentially expressed genes (DEGs) identified by the benchmarked data handling methods (limited to the DEGs that are also identified in full-thickness biopsy).

**Figure 4 ijms-23-06140-f004:**
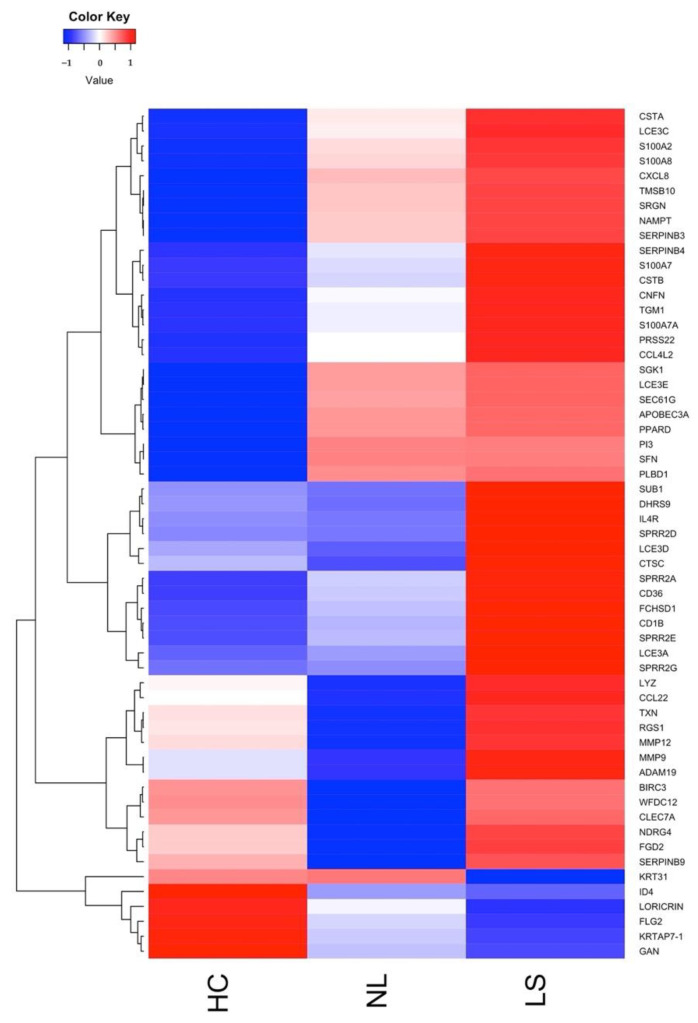
Heatmap and one-way unsupervised hierarchical clustering showing selected AD signature genes obtained from tape-stripped skin samples. Each column represents the Z-scaled group mean gene expression for HC, NL, and LS skin.

**Figure 5 ijms-23-06140-f005:**
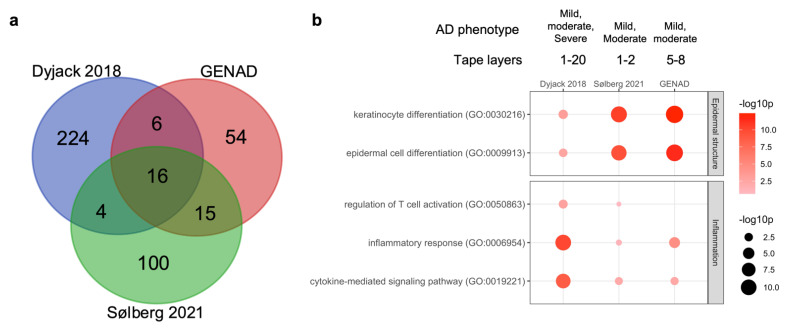
Comparison of transcriptome data from three studies: GENAD, Dyjack 2018 [14], and Sølberg 2021 [13]. (**a**) Venn diagram showing the overlap of AD DEGs; (**b**) bubble plot showing the most enriched pathways. The AD phenotype and tape layers used in the three studies are indicated above the plot.

**Figure 6 ijms-23-06140-f006:**
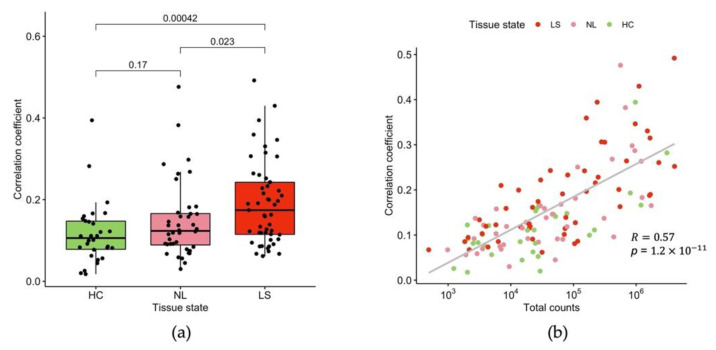
(**a**) Correlation between gene expression detected in tape-stripped skin and full-thickness skin punch biopsies in HC, NL, and LS skin. The *p*-values of the group mean comparison are shown above each contrast; (**b**) The tape strip–punch biopsy correlation increases with the total transcript counts obtained from the tape-stripped skin.

**Figure 7 ijms-23-06140-f007:**
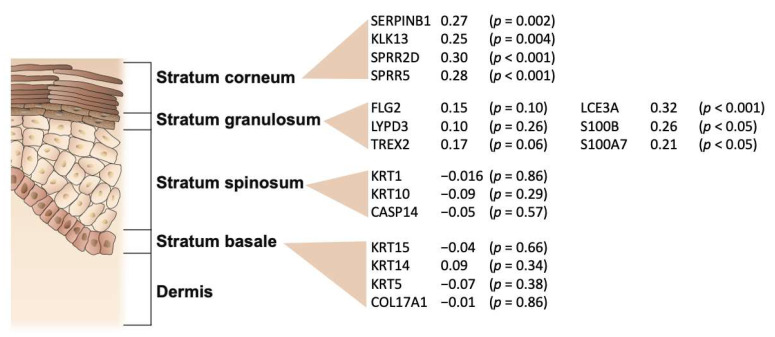
Correlation between gene expression detected in tape-stripped skin and matched full-thickness skin punch biopsies for selected epidermal layer marker genes [16]. Data shown are gene symbol, Spearman correlation coefficient, and *p*-value.

**Table 1 ijms-23-06140-t001:** Benchmark of counts transformation and differential expression (DE) testing methods.

Transformation	DE Testing	DE Cutoff	DEGs (LS vs. NL) Total (Up/Down) ^8^	Accuracy (Up/Down) ^9^
RLE ^1^	edgeR glm fit	FC > 2, padj < 0.05	520 (4/516)	1.15% (0/6)
TMM ^2^	edgeR glm fit	FC > 2, *p* < 0.05	263 (226/37)	8.37% (21/1)
TMMwsp ^3^	edgeR glm fit	FC > 2, *p* < 0.05	242 (196/46)	8.68% (19/2)
ZINB-WaVE ^4^	edgeR glm weighted F	FC > 2, *p* < 0.05	212 (122/90)	5.19% (10/1)
ZINB-WaVE ^4^	DESeq2	FC > 2, *p* < 0.05	78 (56/22)	7.69% (6/0)
Voom-trend ^5^ on TMM data	limma	FC > 2, *p* < 0.05	92 (71/21)	22.83% (21/0)
Voom-trend ^5^ with quality weight on TMM data	limma	FC > 2, *p* < 0.05	168 (151/17)	19.05% (31/1)
Mitochondrial gene set normalization ^6^	edgeR	FC > 2, padj < 0.05	333 (251/82)	8.11% (24/3)
VST ^7^	DESeq2	FC > 2, *p* < 0.05	880 (3/877)	1.02% (0/9)
TMMwsp ^3^	NOIseq	Top-5% ranked	115 (70/45)	16.52% (15/4)
TMMwsp ^3^	NOIseq	Top-20 ranked upregulated	20 (20/0)	55.00% (11/0)
TMMwsp ^3^	NOIseq	Top-10 ranked upregulated	10 (10/0)	70.00% (7/0)

^1^ RLE: relative log expression [18]; ^2^ TMM: trimmed mean of M value [19]; ^3^ TMMwsp: trimmed mean of M value with singleton pairing [19]; ^4^ ZINB-WaVE: Zero-inflated negative binomial modeling [20]; ^5^ voom [21]; ^6^ Mitochondrial genes as stable reference set for normalization [22]; ^7^ VST: Variance stabilization transformation [23]; ^8^ Number of differentially expressed genes (DEGs) identified (upregulated/downregulated); ^9^ Accuracy (number of correctly identified DEGs upregulated/downregulated).

## Data Availability

BRB-seq data on tape-stripped skin samples is available in GEO under accession GSE199046. Full-thickness punch biopsy RNA-seq data are available in GEO under the accession GSE193309.

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
