# Peer review of "Profiling the Atopic Dermatitis Epidermal Transcriptome by Tape Stripping and BRB-seq"

_ijms, 2022, doi:10.3390/ijms23116140_

Round 1

Reviewer 1 Report

The work is methodologically well structured and correlates the increasingly important procedure of tape stripping with classical punch biopsy.
However, there are some minor revisions to be made before final publication.
1) it is necessary to add in the introduction the literature results available to date regarding gene expression found both on biopsies and on tape stripping in atopic dermatitis
2) the discussion should be enriched by contextualising the data with evidence already available in the literature highlighting similarities and differences and providing possible explanations
3) the conclusion section should be separated from the discussion and extended.

Author Response

We have included a point-by-point response in the enclosed document.

Reviewer 2 Report

The authors conducted Bulk RNA Barcoding and sequencing of normal and atopic dermatitis patient skin samples obtained by tape stripping. They confirmed changes in atopic dermatitis signature gene expression in their analyses. In addition, the authors indicated the technical difficulties of Bulk RNA Barcoding and sequencing using tape stripped skin samples.

  1. The authors should provide more information about the advantages and disadvantages of Bulk RNA Barcoding and sequencing in the Introduction.
  1. Lines 220-223, “This may be accomplished by standardizing the sampling conditions, both technical parameters such as controlling the pressure, duration, and speed of tape removal, as well as biological factors, including anatomical site, hydration and stretching of the skin, as proposed by Hughes et al [23].”:

The authors refer to this review article and mention technical problems and difficulties of analyzing RNA sample obtained by tape stripping.

In this current study, the authors controlled the pressure and duration, but they concluded “that deep BRB-seq of tape stripped skin is needed to counteract large between-sample RNA yield variation and highly zero-inflated data in order to apply this protocol for population-wide screening of the epidermal transcriptome in inflammatory skin diseases.”

1) The authors should propose an experimental method to establish standardized methods.

2) The authors reported variation of RNA levels in different anatomical sites in Fig 2.

They should report results of RNA levels in each anatomical site; e.g., leg: NL, LS, HL.

3) Regarding seasonal variation: Is seasonal variation evident in all subjects or in a specific subject group AD (NL or LS)? Why do the authors not report data of Q1-Q4 separately for NL, LS and HL?

  1. Lines 120-123, “We found that no existing data handling method could capture all DEGs. The data handling methods in the benchmark showed and overall low accuracy and low overlap in identified DEGs (Figure 3).”: If KC differentiation abnormalities occur, most of the DEGS expression profile should be altered. It depends upon the aim/goal of the research. But is it necessary to capture all DEGS?
  1. Related comment #1, above, As indicated by the authors, prior articles already have described the technical difficulty of profiling gene expression using tape stripped samples. These authors have a similar conclusion. What is the significance of this current study? The importance of their current study and its contribution to cutaneous research field should be discussed.

Author Response

We have provide a point-by-point response in the enclosed document.

Round 2

Reviewer 2 Report

The authors revised manuscript in response to reviewers’ comments. The manuscript has been improved. This reviewer has some comments.

Lines 177-182

“We compared our AD gene expression signature to those obtained from two other 174 studies where deep RNA-seq was used for profiling the epidermal transcriptome from 175 tape stripped skin [13,14]. As shown in Figure 5a, we found little overlap between AD 176 signature genes from the three studies. In contrast, pathway enrichment analysis showed good agreement between the most enriched pathways, namely those related to epidermal structure and inflammation (Figure 5b). The Dyjack 2018 data showed a stronger inflammatory response, which probably can be ascribed to the more severe AD in that study. "Thus, the limited overlap between AD signature genes could be due to both variation in disease severity and in tape stripping methodology.”

Dyjack et al. analyzed not only severe, but also mild AD patients (non-lesional skin). Is authors’ interpretation “variation in disease severity” justified? 

Is pathway enrichment analysis critical and necessary for Tape Stripping and BRB-seq method?

How can be standardized tape stripping methodology?

Author Response

Dyjack et al. analyzed not only severe, but also mild AD patients (non-lesional skin). Is authors’ interpretation “variation in disease severity” justified? 

Thank you for this comment. As regards the Dyjack et al. data, this comprises 7 mild, 17 moderate, and 6 severe AD patients in addition to healthy controls. The non-lesional skin comes from the same patients as the lesional skin, and thus, cannot be regarded as “mild” cases of AD. If anything, the NL signature may represent the molecular scar in AD patients.

Importantly, the contrast that we have re-analyzed, is lesional vs. healthy controls (Table E1 from Dyjack et al.). Thus, we do believe that the differences between our study and Dyjack et al.’s data are due to both methodological and disease specific variation.

Is pathway enrichment analysis critical and necessary for Tape Stripping and BRB-seq method?

Pathway enrichment analysis may capture trends in the analysis that are not apparent at the gene level only. For example, in those cases, where many small changes in gene expression level may not meet the filtering criteria for single gene analysis, these combined effects may still be able to identify relevant pathways and networks. As such, pathway analysis is not critical for the interpretation of Tape stripped and BRB-sequenced data, but nevertheless, it reflects the high-level patterns in the data, such as inflammatory signatures, and therefore, we believe it is useful for illustrating both similarities and differences between studies.

How can be standardized tape stripping methodology?

We are not sure how to interpret this question, as we believe we have already addressed the issue of standardization of the tape stripping technique in the manuscript. Specifically, we write:

To increase the usability of tape stripping in dermatological research and diagnosis, one could improve the technique, both by minimizing sample variation and by maximizing RNA yield and quality. This may be accomplished by standardizing the sampling conditions, both technical parameters such as controlling the pressure, duration, and speed of tape removal, as well as biological factors, including anatomical site, hydration and stretching of the skin, as proposed by Hughes et al [28]. To avoid RNA degradation during storage, addition of RNA preservatives, such as RNAlater, should be considered. On the other hand, dry storage of non-conserved tape strips at room temperature for up to three days followed by deep sequencing (average depth of 90 M reads per sample) has been used successfully for transcriptomic profiling of the stratum corneum in AD [26] as well as in hand eczema [29].

Thus, we hope that the above discussion is sufficient for the scope of this manuscript with a focus on the molecular characterization of tape stripped skin in AD.

Round 3

Reviewer 2 Report

In method, “More study details can be found in Hu et al [15].”

However, this paper is in press and this reviewer cannot obtain information about experiment. Thus, please provide patient information, i.e., drug or emollient usage and others in method. Since Hu et al study is already in press, this reviewer recommends discuss Hu et al results in comparison to authors’ current study in revision.

  1. Hu, T.; Todberg, T.; Ewald, D.A.; Hoof, I.; Correa Da Rosa, J.; Skov, L.; Litman, T. Assessment of Spatial and Temporal Variation in the Skin Transcriptome of Atopic Dermatitis by Use of Minimally Invasive Punch Biopsies. Journal of Investigative Dermatology 2022, In Press.

Author Response

We have provided the specific patient details in Supplementary Table 1, and replaced the above sentence to read:

“Specific patient details can be found in Supplementary Table 1”.

Also, as privileged information, we have enclosed our manuscript in press for the reviewer.

We hope that this information is now sufficient for our manuscript to be acceptable for publication.

Round 4

Reviewer 2 Report

Authors added information about patients. But ongoing and/or history of drug treatments in supplemental Table. This information is included others' papers cited in the manuscript.